# Genome-Wide Association Study of Growth and Sex Traits Provides Insight into Heritable Mechanisms Underlying Growth Development of *Macrobrachium nipponense* (Oriental River Prawn)

**DOI:** 10.3390/biology12030429

**Published:** 2023-03-10

**Authors:** Mengchao Wang, Shubo Jin, Shuai Liu, Hongtuo Fu, Yunfeng Zhao, Li Jiang

**Affiliations:** 1College of Fisheries and Life, Shanghai Ocean University, Shanghai 201306, China; 2Key Laboratory of Aquatic Genomics, Ministry of Agriculture and Rural Affairs, Beijing Key Laboratory of Fishery Biotechnology, Chinese Academy of Fishery Sciences, Beijing 100141, China; 3Key Laboratory of Freshwater Fisheries and Germplasm Resources Utilization, Ministry of Agriculture, Freshwater Fisheries Research Center, Chinese Academy of Fishery Sciences, Wuxi 214081, China

**Keywords:** *Macrobrachium nipponense*, GWAS, binary trait, heritability, sex trait

## Abstract

**Simple Summary:**

*Macrobrachium nipponense* is an important economic aquaculture animal, and due to its delicious taste, it is a popular food in South China. The growth of male and female *Macrobrachium nipponense* is significantly different. The female grows slowly due to precocious puberty, whereas males have a clear growth advantage. Therefore, due to their fast growth, males are often the subject of genetic breeding by scientists. For this study, we obtained single-nucleotide polymorphisms (SNPs) that could affect growth and sex differentiation to assist artificial selection in genetic breeding. This study entailed a genetic evaluation of 10 growth traits and one sexual trait using genome-wide association analysis (GWAS), while some significant SNPs associated with growth and sexual traits were detected. Moreover, genetic correlations were also found between traits, especially between growth and sexual traits, which greatly simplified genetic selection or made multiple joint selection convenient for more than one trait. At the same time, multiple SNPs were found to be located in two chromosomes and greatly contributed to a high heritability. These results illustrate the genetic nature of growth and sex traits in *Macrobrachium nipponense*.

**Abstract:**

Male hybrid oriental river prawns grow significantly faster than hybrid females. In this study, the growth and sex traits of 181 individuals of *Macrobrachium nipponense* were recorded, and each individual genotype was evaluated using the 2b-RAD sequencing method. The genetic parameters for growth and sex traits were estimated. A genome-wide association analysis (GWAS) of these traits was performed. In total, 18 growth-related SNPs were detected from 12 chromosomes using a mixed linear model. The most significant loci of weight are located on the position of the SNP (102638935, chromosome 13), which can explain 11.87% of the phenotypic variation. A total of 11 significant SNPs were detected on four chromosomes associated with sex trait (three on chromosome 4, one on chromosome 7 and seven on chromosome 17). The heritability of this trait is 0.8998 and belongs to the range of ultra-high heritability. Genetic correlations were prevalent among the 11 traits examined, the genetic coefficient between sex and body weight reached a significant level of −0.23. This study is the first GWAS for sex of binary and growth traits in oriental river prawn. Our results provide a set of markers for the genetic selection of growth traits and help us to further understand the genetic mechanisms of growth in *Macrobrachium nipponense*.

## 1. Introduction

*Macrobrachium nipponense* is a commercial freshwater prawn species, generally known as oriental river prawn and widely distributed worldwide [1]. In aquaculture, the oriental river prawn has great potential to develop due to its strong tolerance of low temperatures in winter and fast growth rate in a typical pond culture environment. It exhibits a sexual size dimorphism starting due to sex differentiation, with male body weight reaching two or three times that of females. However, the body size of females is very small due to early sexual maturation before reaching marketable specifications. As a result, the economic benefits of a mixed culture of *Macrobrachium nipponense,* females and males are greatly reduced due to the growth inhibition of females. Therefore, intensive research should be directed towards the controlled establishment of prawn mono-sex populations [2]. The construction of the all-male or all-female populations via all kinds of physical, chemical or hybrid processing methods is the target of breeding for many species in which a growth dimorphism exists between male and female individuals [3,4]. However, the genetic properties of growth traits in many aquatic species are still unclear. In particular, there are few studies on the genetic correlations between growth traits and between sex and growth traits, which causes many difficulties regarding the improvement of the genetic mechanism of sex and growth traits in aquatic animals.

Research on oriental river prawn mostly focused on the optimization of previous nutrition formulae [5] or the impact of certain nutrients on biochemical and physiological progress [6]. Furthermore, the candidate gene method was used to determine the gene function of similar homologous species in *Macrobrachium nipponense* [7]. An important breakthrough in the genome assembly of *Macrobrachium nipponense* was recently achieved, providing rich genomic data for genetic analysis. 

A genome-wide association study [8] based on a single-nucleotide polymorphism can be conducted to estimate the genetic parameters of every trait and identify genes via the position of SNP on a chromosome. this method is widely applied in animal breeding [9], disease analysis [10] and phenotype prediction [11,12]. In GWAS, according to the features of the phenotypes, phenotypes with category differences, such as diseases, were defined as binary traits [13]. The phenotypes with only numerical differences were defined as continuous traits and other special traits, such as time-to-event traits [14]. Sex, just like disease, is one of the binary traits normally analyzed using a generalized linear model [15] or generalized linear mixed model, which both account for polygene effects [16]. Generalized linear mixed-model association tests (GMMAT) [17], as well as the scalable and accurate implementation of a generalized mixed model (SAIGE) [18] based on generalized linear mixed model, have already become the first choice for the analysis of binary traits. In this paper, a SAIGE was applied to analyze the binary trait of sex since the significance test of GMMAT assumes a Gaussian distribution, which is unsuitable for the analysis of binary traits [19]. Quantitative traits [20], such as body length and its differences, were only measured as numerical values and did not have clear classification features. In this study, we used the efficient mixed-model association expedited (EMMAX) [21] method to analyze all quantitative traits related to growth.

Oriental river prawn is an important aquaculture species native to South China with a huge market and economic benefits. However, few SNPs or quantitative trait loci (QTLs) of growth and sex traits were reported due to the genomic complexity of this species. We first applied genotyping by sequencing in oriental river prawn and used the methods of GWAS, SAIGE and EMMAX to obtain associated SNPs involved in sex and growth traits. The correlation coefficient between traits was evaluated, providing guidance for subsequent breeding programs.

## 2. Materials and Methods

### 2.1. Animal Population and Collection of Phenotype

The populations of the tested river oriental prawn derived from the same family that underwent four generations of successive genetic selection in an aquaculture farm in Xinghua, Jiangsu, China. A total of 81 male and 100 female *Macrobrachium nipponense* individuals with complete phenotype records were randomly selected. Growth traits, including body weight (BW), ration of meat yield (RMY), length of beard (LB), length of hepatopancreas (LH), chest circumference (CC), length of fifth leg (LFL), body length (BL), head width (HW), tail width (TW) and length of pliers (LP) were analyzed. In addition, tissue of each sample from the test population were kept in a cryopreservation tube and placed in a refrigerator at −80 °C for DNA isolation and genotyping by resequencing. 

### 2.2. Isolation of DNA and SNP Calling 

DNA was extracted from the tissues of samples according to a standard phenol–chloroform protocol [22]. A 1% agarose gel was used to evaluate DNA quality, and a spectrophotometer was utilized to measure its concentration. Then, its concentration was adjusted to 2.5 μg/μL. The purified DNA samples were sent to be sequenced.

The quality control for each sample was conducted using an NGS QC Toolkit [23]. Sequence alignment with the reference genome was performed using the software BWA0.4 [24] and SAMtools [25]. The sequences were sorted, and replicates were removed using the software, Picard (https://github.com/broadinstitute/picard accessed on 12 March 2022). The filtering of mutation sites and screening of SNP were carried out. Next, the results were converted to VCF files using GATK software (https://github.com/broadinstitute/gatk accessed on 9 May 2022). Finally, the files of binary traits and other quantitative traits comprising the genotyping dataset were produced using PLINK1.9 software (https://www.cog-genomics.org/plink2/ accessed on 18 July 2022).

### 2.3. Genome-Wide Association Analyses

Classic linear mixed-model association analyses were performed using GEMMA for genome-wide association analyses [26], which treats the phenotype as a fixed factor and the additive polygenic effect as a random effect. The genome-wide, significantly associated SNPs were picked out using a Bonferroni correction, and the heritability of traits was calculated by GEMMA.

A sex trait is a typical binomial trait that does not follow a normal distribution; therefore, a linear mixed model was not suitable for the gene mapping of this trait. For the above reasons, the state-of-the-art method SAIGE [18], based on a generalized linear mixed model, was selected in a genome-wide genetic analysis of sex traits of *Macrobrachium nipponense*. SAIGE is usually used in two steps. The first step is to fit the null logistic mixing model through genomic markers without considering any covariates. In the second step, a univariate association test was carried out for variation data according to the method of leave-one-chromosome-out. 

### 2.4. Statistical Test

Principal component analysis (PCA) using EIGENSOFT was performed to evaluate population structure [27]. The independent SNP dataset was analyzed via a PCA analysis using PLINK. We employed a generalized linear mixed model in our association analysis using SAIGE [28] for the binary trait of sex. The genetic correlation analysis between traits was conducted using the R package PerfomanceAnalytics [29]. 

The Bonferroni-corrected significance threshold (*p* ≤ 0.05) was used to identify the candidate SNPs associated with each trait. The tested SNP was considered as a candidate for a significant SNP when its *p*-value was over the threshold. Significant SNP markers were visualized in a Manhattan plot using Haploview4.2 software [30]. *p*-value distributions (expected vs. observed *p*-values on a −log^10^ scale) are shown in a quantile–quantile plot (Q–Q plot).

## 3. Results

### 3.1. Phenotype and Genotype, SNP Calling and Quality Control 

The descriptive statistics of all the traits are shown in Table 1. The SNP calling of 181 samples was executed using a software package that includes BWA, Bcftools, Samtools, Picards, GATK, etc. After quality control assessment, 7793 SNPs were kept for further genome-wide analysis (Table 2). The significant threshold at the 0.05 level in the multiple tests was 6.42 × 10^−6^ (0.05/7793), and the significance threshold at the 0.01 was 1.28 × 10^−4^ (0.01/7793). A principal components analysis is shown in Appendix A, and the top 10 principal components (*lambda* > 0.5) were used for stratified population correction.

### 3.2. Genome-Wide Association Analysis

A GWAS was performed for these 11 traits (1 binary and 10 quantitative) and evaluated their heritability through methods of GEMMA, respectively (Table 3 and Table 4). The heritabilities of 10 growth traits ranged from 0.16 (BL) to 0.99 (BW, RMY), indicating that most of them had a medium and high heritability.

For the genome-wide association analysis of binary sex trait, three SNPs on chromosome 4, one SNP on chromosome 7, and seven SNPs on chromosome 17 were found. Detailed information on these candidate SNPs is shown in Table 3, and the visual results of GWAS are shown in Figure 1. The heritability of the sex trait was 0.8998, which was significantly higher than the sum of the heritability (0.7577) of all SNPs detected. This result indicates that there was interaction between genes or among multiple SNPs, which were possibly involved in the development of sex traits. Candidate SNPs associated with sex traits were found on the same chromosome with close relative distances and similar significance levels, suggesting that linkage disequilibrium may interfere with the results of GWAS analysis (Table 3). 

For quantitative growth traits, significant SNPs were detected for each trait in the quantile–quantile plot (Q–Q plot) and Manhattan plot (Figure 2). At the same time, no significant SNPs were detected for some growth traits: RMY, LB, CC, LRL, HW, TW, LP (Figure 3, Table 4). In total, 18 SNPs were detected for three growth traits: one SNP was associated with body weight, four were associated with length of hepatopancreas, and thirteen were associated with body length, according to GEMMA-0.98-1 software (Table 4). The positions of candidate SNPs on chromosomes are shown in Table 5.

### 3.3. Correlation Analysis among Traits

A genetic correlation analysis showed that there was a general genetic correlation among these traits (Figure 4), in which sex traits had a significant negative correlation with other growth traits, except LH and BL. Most of the other growth traits had a significant genetic correlation, among which the trait of body weight had a very strong correlation with other growth traits, except LH. The correlation coefficients between BW and others were 0.86 (RMY), 0.71 (LB), 0.76 (CC), 0.74 (FL), 0.33 (BL), 0.90 (HW), TW (0.83) and 0.89 (LP), respectively, indicating a high genetic correlation between body weight and other growth traits.

## 4. Discussion 

Sex is one of the many limitations of efficiency in aquaculture farming [30], and sex differences often lead to large differences in growth performance regarding body weight, body length and growth rate. In general, males have a faster growth rate [2] than females; therefore, breeders use special methods such as sexual reversal to achieve all-male breeding and reduce economic losses caused by the slow growth of females [31]. Thus far, studies related to the sex differentiation [32] of *Macrobrachium nipponense* focused on the cloning and functional analysis of candidate genes [33,34], which have a potential influence on sex determination [35]. However, genetic-associated SNPs across the whole genome remain unclear, which limits the utilization of these genes and selectable markers for breeding. We identified ten significant SNPs associated with sex traits across the genome, mainly located on chromosomes 4 and 17 (Figure 1 and Table 3). The total heritability of the sex trait reached 0.8998, indicating a super high heritability that facilitates easier breeding using the genetic nature of loci associated with sex traits. 

A GWAS of the growth-related traits was carried out using GEMMA, and eighteen SNPs associated with the growth traits were identified across the whole genome. Of these significant SNPs, one is associated with body weight. The estimated heritability of the sex trait reaches 0.99, indicating a super-high heritability. Thirteen SNPs were identified associated with BL and four SNPs associated with LH trait, the estimated heritability of them is 0.45 and 0.16, respectively, they exhibited high and moderate heritability, respectively. The estimated heritabilities of other growth traits were generally high, indicating that the effect of genetic factors for each trait was much greater than that of environmental factors, which is conducive to genetic selection by genome selection and provides theoretical support for the genetic breeding of *Macrobrachium nipponense*. 

This is the first study concerning *Macrobrachium nipponense* to identify SNPs associated with sex and growth traits at a genome-wide level. Our results showed that sex and body weight traits are highly genetically correlated; the genetic correlation between sex and BW trait was 0.23, which indicates a significantly negative correlation, meaning that the detected QTNs can participate in the processes of both sex and growth development. Furthermore, the location of these candidate QTNs can be used for functional gene mapping for target trait control, which is one of the main functions of GWAS. This high correlation between growth traits was similar to the correlations found for Pacific abalone (*Haliotis discus hannai*.) [36]. Interestingly, the genetic correlation between weight and other growth traits was much more positive, which is very suitable for selection breeding in oriental river prawn. The advantage of a high genetic correlation among traits of *Macrobrachium nipponense* can provide a more measurable index and greatly simplify the complexity of genetic selection, which is also used in the selection of the feed conversion of abalone [37]. 

The significant correlation between various growth traits of *Macrobrachium nipponense* showed that, in the online GWAS using GEMMA [37], multiple growth traits can be jointly selected at the whole-genome level, but this was not suitable for binary traits with other growth traits. The reasons for this could be as follows: First, GEMMA is designed for quantitative traits and is not suitable for binary traits, such as sex traits or survival traits. In addition, the calculation rate is greatly reduced when analyzed in combination with growth traits. 

The heritability [38,39] of the traits is the basis of genetic selective breeding. Heritability (mathematically expressed as *h*^2^) is the ratio of genetic variance to total variance. The total variance is also known as the phenotypic variance (the sum of genetic variance and environmental variance). Heritability can also be understood as the influence of genetic factors on traits. The values of heritability calculated using the ratio of genetic to phenotypic variance are not equal to the total variance of all SNPs, which may be due to a number of reasons: firstly, the genetic variance was calculated from all genetic markers, some markers are not significant when tested individually. Secondly, the missing heritability perhaps suggests that the linkage disequilibrium (LD) heterogeneity among regions has an adverse effect on genomic prediction and heritability estimation. As illustrated in our study, the estimated heritability of BW reached 0.99, which was higher than that of the total SNPs (0.91), and such similar cases of “missing” or “hidden” heritability are mentioned in several studies [40,41,42].

On chromosome 17, seven SNPs were found to have significant genetic correlation with sex traits at the whole genome level, indicating that some SNPs had relatively significant effects on chromosome 17. These SNPs enhanced the effects of other detected SNPs that were not significantly correlated and caused them to have a significant genetic correlation. Such a linkage disequilibrium affects the interpretation of GWAS [42]. In addition, there may be interactions between markers that contribute to genetic variance. Since the interaction effect is widespread at the whole genome level, a single step analysis cannot detect such an effect, but it will increase the genetic variance; therefore, the interaction effect often interferes with the analysis of GWAS, which is why a GWAS analysis should be conducted from multiple perspectives. Therefore, improving the detection efficiency and accuracy of GWAS is essential.

In conclusion, this study helps us to further understand the genetic structure of growth and sex traits of oriental river prawn by GWAS. In this study, a total of 11 SNPs related to sex traits were found, suggesting that some genes near these loci may have a significant sex differentiation potential. These candidate genetic markers of SNP may provide valuable resources for further understanding sex determination in oriental river prawn. In addition, 18 SNPs related to growth traits, such as BW, LH and BL, were detected in this study. Interestingly, these SNPs may also affect sex differentiation, based on the results of the correlation relationship between sex and other growth traits. This discovery provides a convenient tool for genetic selection and makes the breeding process more efficient.

## 5. Conclusions

In summary, for the first time, we found some SNPs that were significantly correlated with growth traits and sex traits of oriental river prawn at a genome-wide level. We also found that there was a wide range of high genetic correlations between growth traits and between growth and sex traits. Some growth traits and sex traits of oriental river shrimp not only have a medium and super high heritability, they also demonstrate a high genetic correlation among traits, which provides a more convenient and efficient scheme for genetic breeding, such as multi-trait joint selection. This study provided a basis for understanding the genetic mechanism of the disparity in growth performance between male and female in *Macrobrachium nipponense*. With the publication of detailed genomic data in the future, the genetic markers associated with these traits will be developed into genetic units that can be used for different purposes: genetic selection, identifying genes and their traits, exploring gene networks, and studying their biological functions. Therefore, we can facilitate the genetic selection and improvement of production efficiency for *Macrobrachium nipponense.*

## Figures and Tables

**Figure 1 biology-12-00429-f001:**
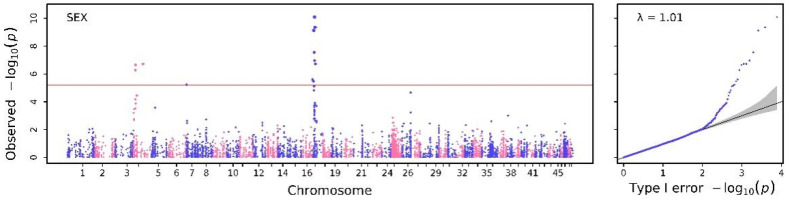
The Manhattan and Q–Q plot for binary trait of sex obtained from SAIGE.

**Figure 2 biology-12-00429-f002:**
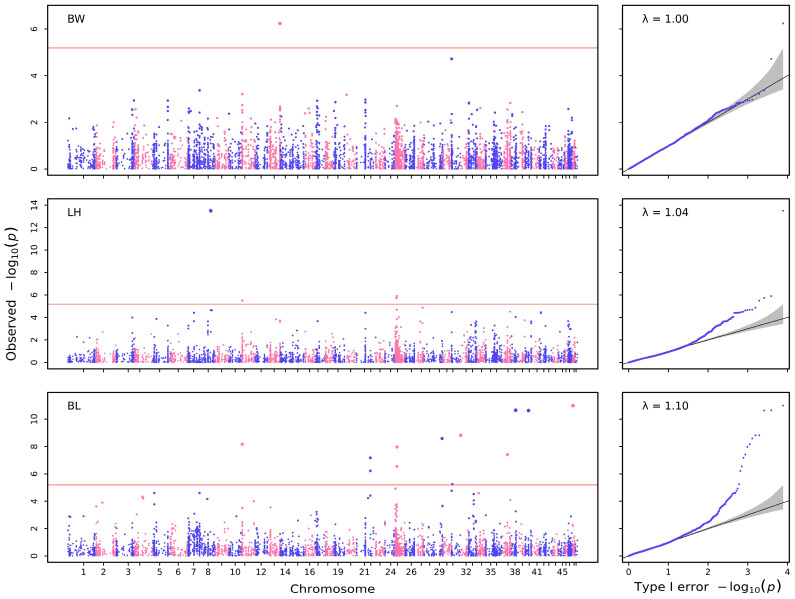
The Manhattan and Q–Q plots for quantitative traits created using GEMMA.

**Figure 3 biology-12-00429-f003:**
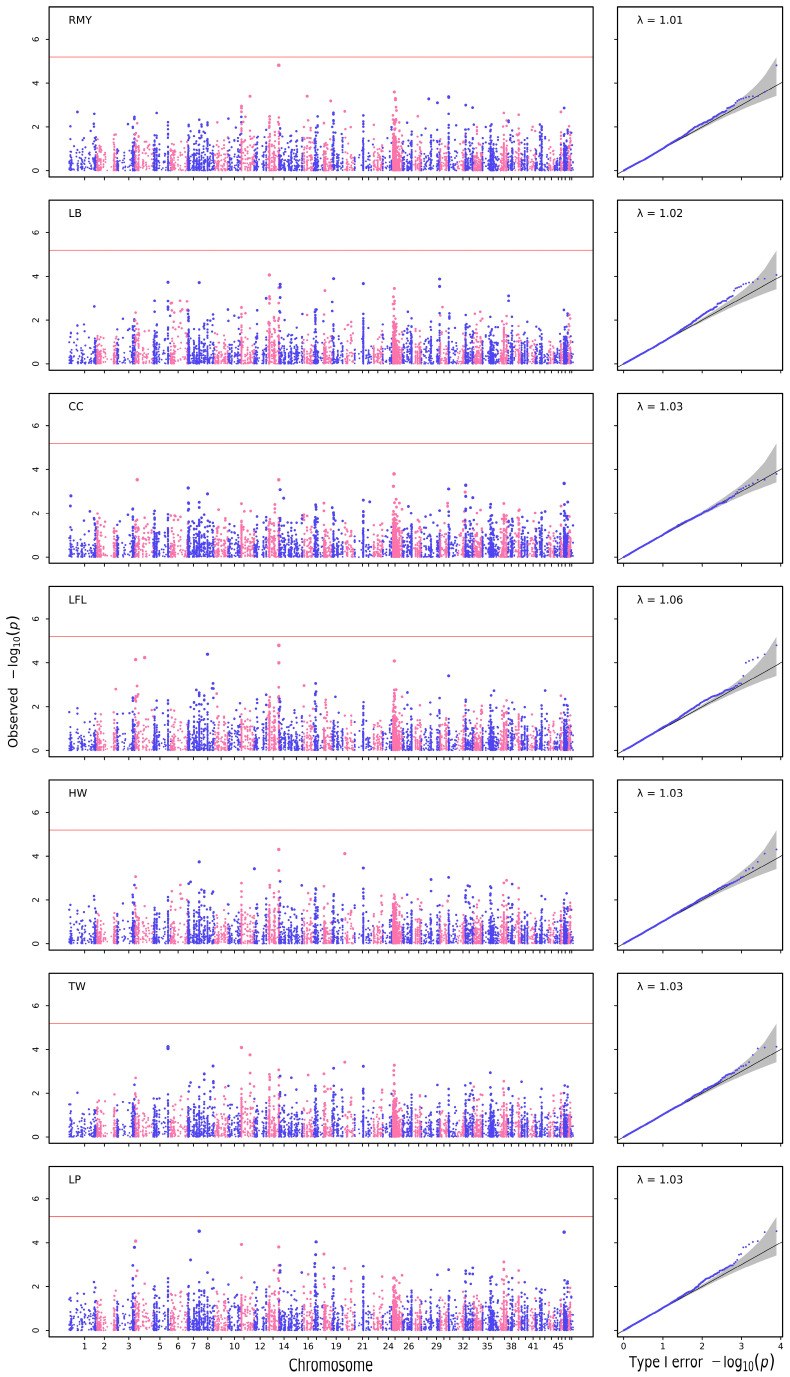
The Manhattan and Q–Q plots for quantitative growth traits created using GEMMA.

**Figure 4 biology-12-00429-f004:**
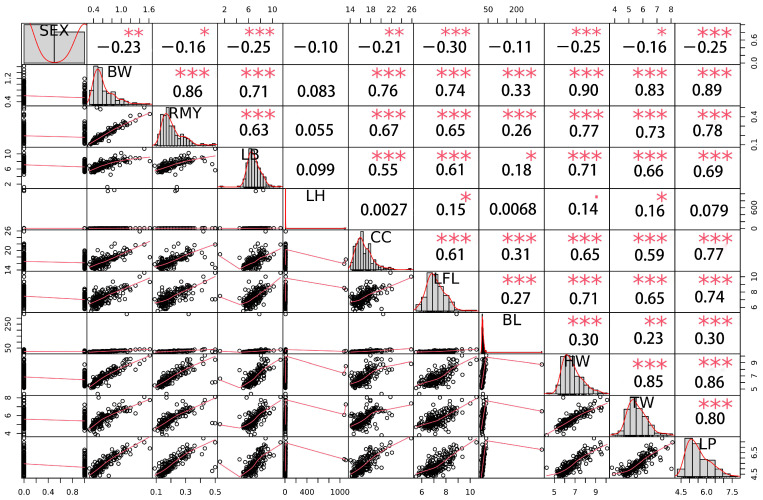
The distribution diagram of genetic correlation among traits obtained using R package PerfomanceAnalytics, the diagonal (the first and fourth quadrants of a pair of traits) is the *p*-value distribution diagram of each trait, the second quadrant is the correlation coefficient and significance level, and the third quadrant is a bivariate scatter plot. In the second quadrant, one star, two stars and three stars represent significant, relatively significant and extremely significant differences at significance levels of 0.05, 0.01 and 0.001, respectively.

**Table 1 biology-12-00429-t001:** Phenotypic means of quantitative traits.

Traits	Mean	S.D.
BW	0.62	0.24
RMY	0.20	0.08
LB	6.89	1.15
LH	9.30	1.50
CC	16.69	1.73
LFL	7.29	0.84
BL	22.09	4.69
HW	6.73	1.02
TW	5.58	0.77
LP	5.38	0.73

**Table 2 biology-12-00429-t002:** The results obtained from SNP calling and quality control.

Items	Before Quality Control	After Quality Control	Numbers of SNP
Female	100	81	
Male	100	100	
Total	200	181	7793

**Table 3 biology-12-00429-t003:** Summary of candidate SNPs detected for the binary trait of sex.

Trait	CHR	Position	Allele1	Allele2	AC_Allele2	AF_Allele2	BETA	−log10^P^	SNP Heritability (Sum = 0.7557)	Trait Heritability
Sex	4	15720532	G	T	60	0.1657	−1.7731	6.2737	0.0702	0.8998
	4	16854185	A	T	38	0.1050	−2.1512	6.6487	0.0774	
	4	77983121	T	C	64	0.1768	−1.8135	6.7215	0.0757	
	7	1196691	C	A	57	0.1575	−1.6184	5.2299	0.0569	
	17	1354721	A	G	19	0.0525	−2.3820	5.6057	0.0537	
	17	8735782	G	A	19	0.0525	−2.3616	5.4769	0.0528	
	17	10806722	C	T	31	0.0856	−2.5689	9.1268	0.0944	
	17	15732561	G	A	25	0.0691	−2.4986	7.5556	0.0749	
	17	17405758	G	A	24	0.0663	−2.4491	6.9668	0.0695	
	17	17406275	C	A	43	0.1188	2.4573	10.0919	0.1102	
	17	21052671	G	A	32	0.0884	−2.5772	9.3536	0.0974	

CHR is the number of the chromosome; position is the site of the SNP on physical map of the genome: allele frequency (AF) and allele count (AC).

**Table 4 biology-12-00429-t004:** Summary of candidate SNPs associated with quantitative growth traits.

Traits	SNPs	Delta	Estimated *h*^2^	Sum_SNP_*h*^2^
BW	1	0.62	0.99	0.91
RMY	0	0.20	0.99	
LB	0	6.89	0.52	
LH	4	21.25	0.45	0.10
CC	0	16.69	0.55	
LFL	0	7.30	0.36	
BL	13	23.79	0.16	0.23
HW	0	6.73	0.71	
TW	0	5.58	0.50	
LP	0	5.38	0.55	

*h*^2^ is heritability obtained by GEMMA.

**Table 5 biology-12-00429-t005:** Details of candidate SNPs detected in quantitative growth traits.

Traits	CHR	Allele1	Allele2	Position	AF	BETA	*p*-Value
BW	13	C	G	102638935	0.017	4.51 × 10^−1^	5.69 × 10^−7^
LH	8	A	G	72682480	0.019	3.14 × 10^2^	3.13 × 10^−14^
	11	T	G	1293551	0.011	2.61 × 10^2^	3.13 × 10^−6^
	25	T	G	11413511	0.011	2.66 × 10^2^	1.85 × 10^−6^
	25	A	C	13704879	0.011	2.70 × 10^2^	1.27 × 10^−6^
BL	11	T	G	1293707	0.014	6.01 × 10^1^	6.76 × 10^−9^
	22	T	G	39519561	0.017	5.17 × 10^1^	6.75 × 10^−8^
	22	A	G	39519603	0.02	4.46 × 10^1^	5.99 × 10^−7^
	25	A	T	15146598	0.017	5.45 × 10^1^	1.07 × 10^−8^
	25	C	T	15146607	0.02	4.58 × 10^1^	2.85 × 10^−7^
	29	T	A	70883628	0.014	6.17 × 10^1^	2.59 × 10^−9^
	31	A	G	6678099	0.023	3.82 × 10^1^	5.71 × 10^−6^
	32	T	A	2639350	0.014	6.25 × 10^1^	1.53 × 10^−9^
	32	A	T	2639352	0.014	6.25 × 10^1^	1.53 × 10^−9^
	37	G	A	40580067	0.017	5.25 × 10^1^	3.93 × 10^−8^
	38	T	C	41076945	0.011	7.60 × 10^1^	2.25 × 10^−11^
	40	T	C	26373891	0.011	7.62 × 10^1^	2.37 × 10^−11^
	48	T	C	5097901	0.011	7.83 × 10^1^	1.02 × 10^−11^

## Data Availability

All data are available within the article and its Appendix A.

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
