# Peer review of "Genome-Wide Association Study of Growth and Sex Traits Provides Insight into Heritable Mechanisms Underlying Growth Development of Macrobrachium nipponense (Oriental River Prawn)"

_biology, 2023, doi:10.3390/biology12030429_

Round 1
Reviewer 1 Report
The aim of this work to identify QTN which affect growth and sex differentiation to assist selective breeding of oriental river prawns has merit and the results presented are encouraging.
However the manuscript is difficult to understand due to the very poor level of English. This manuscript needs to be completely rewritten to be comprehensible and publishable. Some of the worst examples of poor English are listed below, but most sentences throughout contain grammatical errors;
(Section 1) - Contrarily, the female performs potential economic value than male due to body length and weight normally was 2 to 3 times than male and use more little time to get its perk
(Section 2.1) - Totally, one hundred of males and another hundreds of females with full phenotype records were randomly selected.
(Section 2.2) - According to standard phenol-chloroform protocol [22] to accomplish DNA extraction.
(Section 2.3) - The sex is a typical binary trait which not follows normal distribution then its correlation between phenotype and genomic genetic marker cannot be accurately elaborated by linear mixed model.
(Section 2.4) - The Bonferroni corrected thresholds were used to judge the significant level of all SNPs, the tested SNP will be considered as a candidate QTN when its P-value overtops the thresholds. Using Q-Q plot to estimate the degree of the fitted model and detect the exist of false negative or positive errors, then decide if the Manhattan plot is plausible or not.
(Section 3.2) - The heritability of the sex trait was 0.8998 which higher than the summary of all QTN’s heritability, that means the real factor influenced the sex determination may be the interaction effect of multiple SNPs that were not significant in statistic tests.
(Section 4) - Normally, the female one performs more fast growth ability [2], that forced aquaculture farmers apply some special ways such as all-male farming and sex reversal[29] to eliminate the economic loss caused by slow growth performance.
(Section 4) - The results of sex and weight have a significant correlation with other growth traits, that means the QTNs selected from different traits may has the potential to involve the sex differentiation.
(Section 4) – The values of the heritability calculated by ratio of genetic to phenotypic variance do not equal the sum of all QTN’s variance, normally higher or lower than the latter, the reasons may be as the follows: Firstly, the genetic variance was calculated by whole genomic marker, some markers do not perform outstanding when it was tested alone
(Section 5) - The heribitabilities of some of growth traits and sexual trait are medium and super high, so the higher genetic correlations and their own higher heritabilities will help us to have an easier pathway for genetic breeding
Note Section 3.1 commences by stating the descriptive statistics of all traits are shown in Table 1, then describes the contents of this table in words. It is unnecessary to write out the contents of this table.
QTN, QTL and SNP are used interchangeably. Consistent nomenclature should be used.
Author Response
Dear reviewer1,
First of all, I would like to thank you very much for your professional review. I have carefully read every question you have asked and carefully revised it one by one. Every question you have asked is very valuable. I would like to express my heartfelt thanks for uploading the revised draft with marks and the final draft. Please kindly review it again. Also the attachment below is your review comments, so that you can review again with convenience.
Best wishes,
Li Jiang

Reviewer 2 Report
The manuscript presented by Wang et al., describes the results of GWAS studies conducted in Oriental River Prawn to identify QTNs associated with growth and sex traits. The authors opine that the results of this study would contribute to mono-sex culture, in long run. The overall study has clear objectives and deliverables. However, the manuscript requires substantial improvement as detailed below.
I. The presentation requires significant improvement. The usage of language was not appropriate at several instances. For example, reproducing few lines from simple summary and introduction sections of the manuscript below. They indicate contradictory views.
Simple summary:
‘’There are great differences between male and female growth of Macrobrachium nipponense, Females grow slowly due to its feature of precocity, and males have apparent growth advantages.’’
Introduction:
‘’Contrarily, the female performs potential economic value than male due to body length and weight normally was 2 to 3 times than male and use more little time to get its perk.’’
II. The study was conducted on prawns averaging 0.62 g. The specimens are okay for GWAS study involving sex. For growth traits, I have some questions. Generally, the market size of Oriental River Prawn is around 2g. The differences in body weight would be captured efficiently in to genetic markers at higher body weights than at juvenile stage. Is there any reason for choosing juvenile prawn for this study.
III. The methods section is not complete. For example, the details of RAD sequencing were not provided. All the steps between ‘samples sent to sequence’ and ‘the quality control for each sample….’ are to be provided in the manuscript. The sequence datasets could not be checked during review process. The datasets should have been uploaded to a public repository and their identifiers are to be provided. Alternatively, they should have been hosted in a server and reviewers would be given access to check them. Few more details of SNPs could have been given.
With these constraints, the manuscript needs revision.

Author Response
Dear Reviewer2,
First of all, I would like to thank you very much for your professional review. I have carefully read every question you have asked and carefully revised it one by one. Every question you have asked is very valuable. I would like to express my heartfelt thanks for uploading the revised draft with marks and the final draft. Please kindly review it again. Also the attachment below is your review comments, so that you can review again with convenience.
Please see the attachment.

Reviewer 3 Report
Reviewer’s comments
Title: Genome-wide association study of growth and sex traits provides insight into heritable mechanisms underlying growth development of Macrobrachium nipponense (Oriental River Prawn)
Manuscript Number: 1997912
Journal: Biology
The research work entitled “Genome-wide association study of growth and sex traits provides insight into heritable mechanisms underlying growth development of Macrobrachium nipponense (Oriental River Prawn)” presents research on single nucleotide polymorphism (SNP) dependency association of growth phenotype (e.g., body wight, length of hepatopancreas, and body length) and sex traits of M. nipponense, an important economic aquaculture animal. This work has employed various bioinformatics tool to reveal SNP location on chromosome, which made a holistic showcase in genome database bioinformatics. However, there are several concepts and diagrams proposed by the author in the article that needs to be properly modified. I still believe that this work can provide a progressive contribution to the accurate prediction of SNPs function and annotation of hypothetical genes in genomic databases of M. nipponense. I suggested this work may be “major revision” for publication in “Biology”. Specific comments and general comments are given below:
Specific comments
1. Too many formatting errors are found in the manuscript, and English is recommended for further editing and proofreading. Some are listed in General comments.
2. There are many abbreviations for proper nouns in the manuscript, and it is recommended to write the full name for the first time. For example, quantitative trait nucleotides (QTNs), quantitative trait locus (QTL), generalized linear mixed model association tests (GMMAT), scalable and accurate implementation of generalized mixed model (SAIGE), efficient mixed-model association expedited (EMMAX), quantile-quantile plot (Q-Q plot), Allele Frequency (AF), and allele count (AC).
3. In Table1, the purpose of this work is to define the growth-related SNPs in the sex binary traits of oriental river prawn. It is recommended that the authors divide all data into male and female to highlight the difference in growth-related traits between males and females.
4. All the Figures are blurred, especially Figure 2, I recommend the author to re-upload the images with the higher resolution.
5. In Figure 4, this is a relatively complex diagram. I suggest that the authors label the titles of the four quadrants. The meaning of the numbers and symbols should be annotated in the figure caption. Considering that the author uses * to distinguish the magnitude of statistical values, the font size of statistical figures is recommended to be uniform.
General comments
1. The semantic and grammatical expressions of the following sentences are incorrect or unclear, and the authors are advised to rewrite them.
l Line 1 in Abstract- There is a big difference … female individuals.
l Line 8 in Abstract- For sexual trait, 11 QTNs… super high heritability.
l Last line of the third paragraph in Introduction- The method of EMMAX was…quantitative traits related to growth.
l First line of the second paragraph in 2.4. Statistical test-The Bonferroni corrected thresholds were used to judge the significant level of all SNPs, the tested SNP will be … overtops the thresholds.
2. Line 8 in Abstract- choromosome4, à chromosome 4 (typo and blank missing); chromosome17 à chromosome 17 (blank missing)
3. Line 6 of the first paragraph in Introduction- …promotion of Macrobrachium nipponense à …promotion of Macrobrachium nipponense. (dot missing)
4. Line 4 of the fourth paragraph in Introduction-...the method of GWAS, SAIGE and EMMAX to…à …the method of GWAS, SAIGE, and EMMAX to… (comma missing)
5. Line 3 in 2.2. Isolation of DNA and SNP calling- μg/μl à μg/μl (fonts are inconsistent)
6. 2.3. Genome-Wide association analyses à 2.3. Genome-wide association analyses (lowercase)
7. Line 1 in 2.4. Statistical test-(Price et al., 2006) à Please cite a reference in a formal format and put in the list.
8. Title of Table 3, QTNS à QTNs (lowercase)
9. Line 4 of the sixth paragraph in Discussion-Linkage Disequilibrium à linkage disequilibrium

Author Response
Dear reviewer3,
First of all, I would like to thank you very much for your professional review. I have carefully read every question you have asked and carefully revised it one by one. Every question you have asked is very valuable. I would like to express my heartfelt thanks for uploading the revised final draft. Please kindly review it again. Please see the attachment.

Round 2
Reviewer 1 Report
The level of English has improved sufficiently to comprehend the manuscript but simple grammatical issues remain throughout such as incorrect usage of singular and plural (e.g. males of hybrid oriental river prawn grows significantly faster than hybrid females), spelling (e.g. But its body size of the female is very small due to early-maturing sexually before reaching to the maketable specifications) and tense (e.g. At the same time, there is no significant SNP was detected for some growth traits such as RMY, LB, CC, LRL, HW, TW, LP (Figure 3, Table 4). To correct these issues, the manuscript would need to be reviewed by an English editing service.
Author Response
Dear Reviewer,
Your scientific advice is very pertinent. I took your advice and carefully revised the whole text especially in extensive editing of English language and style. Please refer to it.
Best Wishes,
Li Jiang
